# Epigenetic Treatment Alters Immune-Related Gene Signatures to Increase the Sensitivity of Anti PD-L1 Drugs

**DOI:** 10.3390/cancers17152431

**Published:** 2025-07-23

**Authors:** Chonji Fukumoto, Pritam Sadhukhan, Masahiro Shibata, Muhammed T. Ugurlu, Rachel Goldberg, David Sidransky, Luigi Marchionni, Fenna C. M. Sillé, Mohammad Obaidul Hoque

**Affiliations:** 1Department of Otolaryngology-Head and Neck Surgery, Johns Hopkins University School of Medicine, Baltimore, MD 21231, USA; chonji-f@dokkyomed.ac.jp (C.F.); pritamsadhukhan.bi@gmail.com (P.S.); shibatamasahiro1124@yahoo.co.jp (M.S.); mtalhaugurlu@gmail.com (M.T.U.); goldberg.rachelr@gmail.com (R.G.); dsidrans@jhmi.edu (D.S.); 2Department of Oral and Maxillofacial Surgery, Dokkyo Medical University School of Medicine, Tochigi 321-0953, Japan; 3Medical Support Center, Ichinomiyanishi Hospital, Ichinomiya 494-0001, Japan; 4Department of Hematology & Oncology, SUNY Downstate Health Sciences University, Brooklyn, NY 11203, USA; 5Department of Oncology, Johns Hopkins University School of Medicine, Baltimore, MD 21231, USA; 6Department of Pathology and Laboratory Medicine, Weill Cornell Medicine, New York, NY 10021, USA; lum4003@med.cornell.edu; 7Department of Environmental Health & Engineering, Johns Hopkins Bloomberg School of Public Health, Johns Hopkins University, Baltimore, MD 21205, USA; fsille1@jhu.edu; 8Department of Urology, Johns Hopkins University School of Medicine, Baltimore, MD 21231, USA

**Keywords:** head and neck squamous cell carcinoma, epigenetic treatment, STAT1, STAT3, PD-L1

## Abstract

Head and neck squamous cell carcinoma (HNSCC) is a common cancer with poor outcomes despite intensive treatment. While immunotherapy using immune checkpoint inhibitors offers new hope, its effectiveness remains limited. In this study, we experimentally examined whether combining epigenetic drugs with anti-PD-L1 therapy could enhance treatment efficacy. Epigenetic treatment changed the expression of several immune-related genes, including increased levels of STAT1, STAT3, and PD-L1. In a syngeneic mouse model, the combination of epigenetic drugs and anti-PD-L1 antibodies significantly reduced tumor growth. Tumors treated with epigenetic drugs showed significantly higher expression of STAT1, STAT3, and PD-L1 compared to untreated tumors. The increased expression of PD-L1 is presumed to enhance the effectiveness of anti-PD-L1 therapy. These results suggest that epigenetic treatment can reprogram tumors to be more responsive to immunotherapy, highlighting a promising new strategy for improving outcomes in patients with HNSCC.

## 1. Introduction

Head and neck squamous cell carcinoma (HNSCC) is the seventh most prevalent cancer worldwide, and despite advances in treatment options such as surgery, radiation, and various anti-cancerous drugs, the five-year survival rate for patients with HNSCC is unfavorable [1,2,3,4,5]. Based on human papillomavirus (HPV) status, HNSCC is classified into two subclasses: HPV-positive and -negative. The HPV-negative subclass is strongly associated with smoking and alcohol consumption, and patients in this group generally exhibit higher resistance to treatment, particularly chemotherapy and radiotherapy, resulting in poorer clinical outcomes compared to HPV-positive patients [6]. Recently, interest in the utilization of immuno-oncologic (IO) agents for advanced HNSCC has increased, including the Food and Drug Administration (FDA) approval of pembrolizumab and nivolumab for patients with recurrent or metastatic HNSCC [6]. However, the overall response rate of these drugs is around 20%, which means that 80% of patients remain unresponsive to these IO agents [7,8,9,10].

Recently, numerous efforts have aimed to enhance the sensitivity of IO agents. One potential strategy involves the combination of IO agents with epigenetic treatment [11,12]. Several studies have reported that epigenetic alterations in tumor cells can reshape gene expression profiles in a manner that facilitates immune evasion [11,12,13,14]. Recently, another study has shown that DNA methyltransferase inhibitors enhance immune signaling via the viral defense pathway [15]. Accordingly, the epigenetic drugs induced type I interferon (IFN)-related genes, such as IFIs, IFNβ1, IRF7, OASL, and STAT1, in ovarian cancer cells and sensitized them to anti-CTLA4 immunotherapy in a melanoma mouse model [13,15]. Type I IFN, which includes IFNα and IFNβ, leads the tumor attraction of lymphocytes to activate the host’s innate immune-sensing pathways [16,17]. Considering these notions, epigenetic treatment might enable us to enhance the efficacy of IO agents through the stimulation of type I IFN-related genes. However, how epigenetic treatment influences the efficacy of immunotherapy in HNSCC is unclear. In this study, we sought to clarify the changes in immune-related gene signatures under epigenetic treatment. Moreover, we explored whether epigenetic drugs increase the efficacy of anti PD-L1 drugs in HNSCC.

## 2. Materials and Methods

### 2.1. Cell Lines

Three human papillomavirus (HPV)-negative HNSCC cell lines (JHU011, JHU022 and UMSCC22b) and three HPV-positive HNSCC cell lines (UMSCC047, UPCISCC090 and 93VU147T) were used in this study. JHU011 and JHU022 were developed from HNSCC in the Division of Head and Neck Cancer Research at Johns Hopkins University (Baltimore, MD, USA) from recurrent larynx cancer (JHU011) and lymph node metastasis from tonsil cancer (JHU022). These cell lines were cultured in RPMI-1640 medium (Mediatech, Manassas, VA, USA) supplemented with 10% fetal bovine serum and 1% Penicillin Streptomycin Solution (Mediatech). UMSCC22b and UMSCC047 were provided by Dr. Thomas Carey (University of Michigan, Ann Arbor, MI, USA) from lymph node metastasis from hypopharynx cancer (UMSCC22b) and from tongue cancer (UMSCC047). UPCISCC090 and 93VU147T were provided by Dr. Susanne Gollin (University of Pittsburgh, Pittsburgh, PA, USA) and Dr. Johan de Winter (VU University Medical Center, Amsterdam, The Netherlands), respectively, from recurrent base-of-tongue cancer (UPCISCC090) and floor-of-mouth cancer (93VU147T). These cell lines were cultured in DMEM high-glucose medium (Mediatech) supplemented with 10% fetal bovine serum and 1% Penicillin Streptomycin Solution (Mediatech). All cells were stored in the cell bank at Johns Hopkins University, Department of Otolaryngology and Head & Neck Surgery, and incubated at 37 °C in an atmosphere of 5% CO_2_.

### 2.2. Epigenetic Treatment

Cell lines were treated with 5-azacytidine (5-aza; Sigma-Aldrich, St. Louis, MO, USA; #A2385), that is, a DNA methyltransferase inhibitor, and romidepsin (Selleckchem, Houston, TX, USA; #S3020), that is, a histone deacetylase (HDAC) inhibitor. Each cell line was seeded at a low density in its respective culture medium and maintained for 24 h before treatment with 500 nM 5-aza (stock solution dissolved in phosphate-buffered saline (PBS)). Cells were exposed to 5-aza for 72 h and differing concentrations of romidepsin (stock solution dissolved in dimethylsulphoxide (DMSO)) for the final 24 h. Controls were handled similarly and treated with vehicle. Cells were harvested 24 h after the last day of treatment for subsequent RNA extraction.

### 2.3. Cell Viability Assay

Optimal numbers of the respective cells were seeded into each well of a 24-well tissue culture plate 1 day before treatment with romidepsin and treated with various concentrations of romidepsin for 24 and 48 h. Fresh Opti-MEM Reduced Serum Medium (Thermo Fisher Scientific, Waltham, MA, USA) was added to the plates prior to analysis. Alamar blue (Bio-Rad, Hercules, CA, USA) was added to the Opti-MEM to a final concentration of 10% and incubated for 1 h. The fluorescence at 560 nm was recorded using a Spectra Max 250 plate reader (Molecular devices, Sunnyvale, CA, USA). Each assay was performed in triplicate, and each experiment was repeated at least three times. Data represents the percentage of viable cells (as compared to vehicle treatment) after 24 and 48 h treatment with varying concentrations of romidepsin. IC50s for the romidepsin treatment of each cell line were determined.

### 2.4. RNA Isolation and cDNA Synthesis

Sample quality assessment and microarray analysis were performed at Sidney Kimmel Cancer Center Microarray Core Facility at Johns Hopkins University, supported by NIH grant P30 CA006973, entitled Regional Oncology Research Center. Total RNA was extracted using the RNeasy Mini Kit (Qiagen, Valencia, CA, USA) with an on-column DNase I digestion step to eliminate genomic DNA contamination. The purity and integrity of the RNA were confirmed by measuring OD260/280 and OD260/230 ratios with a Nanodrop-1000 spectrophotometer and analyzing samples with a Bioanalyzer (Agilent Technologies, Santa Clara, CA, USA). cDNA was synthesized from 1 μg total RNA using M-MLV Reverse Transcriptase (Thermo Fisher Scientific) and a random primer (Sigma-Aldrich).

### 2.5. Microarray Array

To examine the global gene expression of the samples, HumanHT-12 v4 Expression BeadChip arrays (Illumina, San Diego, CA, USA) were applied for microarray hybridization experiments. This platform covers over 25,000 annotated genes using 47,323 unique probes sourced from the National Center for Biotechnology Information Reference Sequence (NCBI RefSeq; Build 36.2) and UniGene (Build 199) databases.

Briefly, 500 ng of total RNA from each sample was amplified and labeled with the Illumina TotalPrep RNA Amplification Kit (AMIL1791, Ambion, Austin, TX, USA) according to the manufacturer’s instructions. For hybridization, 750 ng of biotin-labeled cRNA was mixed with hybridization buffer and incubated on the array at 58 °C for 16–20 h. Following hybridization, the cartridge was opened, and arrays were washed with buffer at 55 °C and then blocked at room temperature. The bound biotinylated cRNA was stained with streptavidin-Cy3 and subsequently washed. Dried arrays were kept in a light-protected box until scanning with the iScan System. Data were extracted using the Gene Expression Module in GenomeStudio Software 2011.1. Background-subtracted probe-level data exported from GenomeStudio were analyzed using Agilent’s GeneSpring GX v12.6. Briefly, the data were first log2-transfomed and summarized to gene-level based on gene symbols. To reduce the false positive rate in later fold-change calculation, data points with signal intensities lower than 10 were floated to 10. Differentially expressed gene targets were selected by performing a moderated t-test (no correction, *p* < 0.05 and fold change > 1.5) between two groups (epigenetic-treated cells and untreated cells), with each group having at least 3 samples. For groups with only one sample, differentially expressed gene targets were selected by fold change only (fold change > 2.0).

### 2.6. TaqMan Low-Density Human Immune Array

Gene expression profiling was performed using the TaqMan Array Human Immune Panel (Thermo Fisher Scientific, Waltham, MA, USA, #4370573) according to the manufacturer’s instructions. This panel tests the expressions of 90 immune function genes and 6 housekeeping genes on a 384-plex genecard (Applied Biosystems, Carlsbad, CA, USA). First, cDNA samples from two groups (epigenetic-treated cells and untreated cells) equivalent to 75 ng of RNA (50 μL) were mixed with an equal volume of TaqMan Gene Expression Master Mix (Applied Biosystems), centrifuged, and loaded on the port on a TLDA card (Applied Biosystems). The genecard was sealed and PCR amplification was performed for 40 cycles of 2 min at 50 °C, 10 min at 94.5 °C, 30 s at 97 °C, and 1 min at 59.7 °C using a Quantstudio 12 K Flex system (Applied Biosystems). Relative gene expression values were obtained employing the comparative Ct method using Expression Suite Software v1.1. 18s RNA was used as an endogenous control.

### 2.7. IFN Response TaqMan Custom Card

mRNA expression levels of type I IFN-related genes, including *DDX41*, *DDX58*, *IFI16*, *IFI27*, *IFI44*, *IFI44L*, *IFI6*, *IFNB1*, *IRF7*, *MB21D1*, *MX1*, *OASL*, *STAT1*, and *TMEM173* [13,17], between the two groups (epigenetic-treated cells and untreated cells) were evaluated using Custom Taqman Gene Expression Array Cards (Life Technologies, Carlsbad, CA, USA) and a Quantstudio 12K Flex system (Applied Biosystems). Relative gene expression values were obtained employing the comparative Ct method using Expression Suite Software v1.1. 18S RNA was used as an endogenous control.

### 2.8. Quantitative Real-Time Reverse-Transcription Polymerase Chain Reaction (qRT-PCR)

Each targeted gene was amplified with a TaqMan Gene Expression Assay (Thermo Fisher Scientific). 18S was amplified as an endogenous control. qRT-PCR between two groups (epigenetic-treated cells and untreated cells) was conducted with TaqMan Fast Advanced Master Mix (Thermo Fisher Scientific) in triplicate using 7900HT FAST Real-Time PCR System (Thermo Fisher Scientific). The expression level of each gene of interest was quantified by 2-ΔΔCT method [18]. Each sample was tested three times.

### 2.9. In Vivo Homograft Therapeutic Experiments

For therapeutic experiments in syngeneic mouse model, 4–6-week-old C3H/HeJ mice were purchased from Charles River Laboratories (Wilmington, MA, USA). SCCVII (3.5 × 10^5^ cells), a murine HNSCC cell line, suspended in 100 μL of PBS was injected into both flanks of mice (Figure 1). Tumor volume was measured every three days and calculated by the following formula: (volume) = (larger diameter) × (smaller diameter)^2^ × 1/2 [19]. When tumor volumes reached 100 mm^3^, mice were randomly assigned to each therapeutic group (seven mice per group). However, the sides of tumors that did not form or were less than 100 mm^3^ were excluded. 5-aza, romidepsin, and anti PD-L1 antibody (BioXCell, West Lebanon, NH, USA) were used for the treatment. 5-aza (0.2 mg/kg) and romidepsin (2 mg/kg) were administered via intraperitoneal (i.p.) injection three times a week [13,20]. 5-aza was diluted in 100 μL normal saline, and romidepsin dissolved in DMSO was diluted in 100 μL PBS. PD-L1 Ab (200 μg/mouse) mixed with InVivo Pure pH 6.5 Dilution Buffer (BioXCell) was administered via i.p. injection twice a week [21]. Control mice were injected with the same volume of PBS, 3% DMSO in normal saline, and InVivo Pure pH 6.5 Dilution Buffer. Therapeutic efficacy was evaluated from the percentage change in tumor volume compared to the tumor volume before treatment was started [19]. These experiments were approved by Johns Hopkins University Animal Care and Use Committee (#MO17M144), and mice were maintained in accordance with the American Association of Laboratory Animal Care guidelines.

### 2.10. Western Blotting

Total proteins from resected tumors were extracted using RIPA buffer (Thermo Fisher Scientific) supplemented with proteinase inhibitor and phosphatase inhibitor cocktails (Roche, Mannheim, Germany). Thereafter, total protein lysate (50 μg) was electrophoretically transferred onto polyvinylidene difluoride membrane; the membrane was blocked with 5% skim milk in 0.05% tris-buffered saline with 0.1% tween and incubated at 4 °C overnight with each protein’s primary antibody. Then, the membrane was probed with an appropriate secondary antibody (Cell Signaling Technology, Danvers, USA). β-actin served as an endogenous control. The primary antibodies and dilution ratios used here are as follows: β-actin (Cell Signaling Technology, Danvers, MA, USA; #8457; 1:1000), PD-L1 (EPR20529; Abcam, Cambridge, UK; #213480; 1:1000), STAT1 (Cell Signaling Technology; #9172; 1:1000), and STAT3 (124H6; Cell Signaling Technology; #9139; 1:2000). Densitometry readings of Western blotting bands were quantified using ImageJ version 1.54p [23]. Each band was measured three times. The obtained values were used to calculate the relative band intensity that was normalized by β-actin.

### 2.11. Statistical Analysis

A Kruskal–Wallis test with a post hoc test (Steel-Dwass test) was used for comparing multiple groups in the mouse experiments. The estimated variation is indicated in each graph as standard error of mean (SEM). These analyses were performed on JMP 12 software (SAS Institute, Cary, NC, USA), and *p* < 0.05 was considered significant.

## 3. Results

### 3.1. Alterations of Immune-Related Signatures by Epigenetic Treatment In Vitro

To investigate overall immune profile changes by epigenetic treatment in HNSCC, we treated HNSCC cell lines with a combination of 5-aza and romidepsin. Three HPV-positive HNSCC cell lines (JHU011, JHU022, and UMSCC22b) and three HPV-negative HNSCC cell lines (UMSCC047, UPCISCC090, and 93VU147T) were treated with a low dose of 5-aza (500 nmM). Doses and duration for romidepsin treatment were determined individually for each cell line based on Alamar blue cell viability assay (Appendix A). Microarray analysis demonstrated a number of genes significantly changed by treatment with 5-aza and romidepsin (Appendix A).

To validate the microarray findings and further investigate immune profile changes following the epigenetic treatment, we utilized a TaqMan Low Density Human Immune Array to evaluate gene expression changes of 90 target immune-related genes and 6 housekeeping genes as a loading control. Genes whose expression levels were altered (either upregulated or downregulated) by 1.5-fold or greater in four or more of the cell lines are listed in Appendix A. Among 90 immune-related genes, the expression levels of *HLA-DRA*, *HMOX1*, *IL12A*, *NFKB2*, *RPL3L*, and *STAT3* were found to be upregulated, and those of *CSF1*, *CSF2*, and *FAS* were downregulated in epigenetically treated cells compared to untreated cells (Table 1). To evaluate these alterations more precisely, these gene expression levels were determined with qRT-PCR using the same primer assays as those found in the array. Compared to untreated cells, in epigenetically treated cells, the expression level of *HLA-DRA* increased more than 1.5-fold in all six cell lines. The expression levels of *HMOX1* and *RPL3L* were increased in five cell lines, except for UMSCC22b. The expression levels of *IL12A*, *STAT3*, and *NFKB2* were increased in four, three, and two cell lines, respectively (Figure 2A). On the other hand, the expression levels of *CSF1* and *CSF2* decreased more than 1.5-fold in five cell lines, and *FAS* decreased in four cell lines after epigenetic treatment (Figure 2B).

One group recently reported an upregulation of IFN-responsive genes following treatment with demethylating agents in ovarian cancer cells [13]. Therefore, we designed an IFN Response TaqMan custom card to evaluate alterations in the mRNA expression levels of fourteen type I IFN-related genes after epigenetic treatment. Among the 14 genes, *IFI6*, *IRF7*, and *STAT1* were upregulated in four or more of the cell lines following the treatment with 5-aza and romidepsin, while the expression of *OASL* was upregulated in two cell lines and downregulated in four cell lines (Table 2). When these genes’ expression levels were evaluated with qRT-PCR, *IFI6* was upregulated in four cell lines, whereas the expression levels of *IRF7* and *STAT1* were increased in three cell lines (Figure 3A). The alteration of the *OASL* expression levels was consistent with the result of the custom card (Figure 3B). Furthermore, we evaluated the mRNA expression levels of *PD-L1*, that is, a key molecule that contributes to promoting an immunosuppressive tumor microenvironment (TME), which was not included in the former two arrays [24]. Furthermore, the *PD-L1* expression level was found to be increased in four cell lines after epigenetic treatment (Figure 3C).

Collectively, these results indicate that epigenetic treatment induces significant changes in immune-related genes, thereby altering the overall immune profile of HNSCC cell lines in vitro.

### 3.2. Therapeutic Impacts of Epigenetic Treatment and Anti PD-L1 Drug in a Syngeneic Mouse Model

Since several immune-related signatures, including PD-L1, were changed after the epigenetic treatment in vitro, we explored the therapeutic efficacy of the combinational treatment of epigenetic drugs and the anti PD-L1 drug in a syngeneic mouse model (Figure 1).

SCCVII cells, a murine HNSCC cell line, were injected into both flanks of C3H/HeJ mice. When tumors grew to 100 cm^3^, mice were randomly assigned to four treatment groups, which were named ‘Control’, ‘Epigenetic’, which indicates 5-aza and romidepsin, ‘Anti-PD-L1’, which indicates the anti PD-L1 antibody, and ‘Combination’, which indicates 5-aza, romidepsin, and the anti PD-L1 antibody (seven mice per each group). As a result, ‘Combination’ showed significantly slower growth compared to ‘Control’ (*p* = 0.012, while ‘Epigenetic’ and ‘Anti PD-L1′ did not show statistical differences compared to ‘Control’ (*p* = 0.058 and *p* = 0.110, respectively; Figure 4A).

We evaluated the protein expressions of STAT1, STAT3, and PD-L1 in the treated tumors. STAT1 expression was increased in tumors in the ‘Epigenetic’ and ‘Combination’ groups, but not in those in the ‘Control’ and ‘Anti PD-L1′ groups (Figure 4B and Appendix A), and STAT3 expression was increased in tumors in the ‘Epigenetic’, ‘PD-L1’, and ‘Combination’ groups, but not in those in the ‘Control’ group (Figure 4B and Appendix A). These results suggest that epigenetic treatment raised these proteins’ expression levels, which is consistent with our in vitro data (Figure 3). In addition, PD-L1 was increased in tumors in the ‘Epigenetic’ group. PD-L1 expression in tumors in the ‘Combination’ group was higher than that in the ‘Anti PD-L1′ group and lower compared to the ‘Epigenetic’ group (Figure 4B and Appendix A). These results indicate that epigenetic therapy increases PD-L1 as well as STAT1 and STAT3, making tumors more susceptible to anti-PD-L1 drugs.

In summary, treatment with 5-aza and romidepsin upregulates STAT1, STAT3, and PD-L1, potentially resulting in an increased therapeutic efficacy of the anti PD-L1 drug.

## 4. Discussion

Most patients with HNSCC are diagnosed at advanced stages, making them more susceptible to recurrence and metastasis, which contributes to a poor prognosis. Accumulating evidence suggests that immunotherapy leads to a better prognosis in patients with recurrent and/or metastatic (R/M) HNSCC than widely used standard care of treatments like platinum-based chemotherapy or cetuximab [7,25]. Recently, immune checkpoint inhibitor (ICI) monotherapy and combinations of ICIs and chemotherapy have been approved as first- and second-line regimens for this group of patients [7]. However, the therapeutic efficacy of these regimens is limited, particularly for patients who cannot tolerate the combination of immunotherapy with chemotherapy [26]. On the other hand, about 60% of patients with recurrent or metastatic HNSCC are non-responders to immunotherapy [27]. Due to limited preclinical and clinical studies, treatment options after progression on immunotherapy are not well established in current guidelines. Therefore, there is a crying need to perform more preclinical and clinical studies of therapeutic agents as monotherapy and in combination with immunotherapy to improve the outcomes of treatment and to overcome immune resistance [28].

In this study, with the intention of understanding the epigenetic alterations following ICI response, we first examined the alterations of gene signatures by treating HNSCC cell lines with epigenetic agents. Using different high-throughput methods and gold-standard qRT-PCR, we determined the alterations of several immune-related genes, including STAT1, STAT3, and PD-L1, after treatment of the HNSCC cell lines with epigenetic agents. To investigate the therapeutic efficacy of the combination of epigenetic and ICI drugs (anti PD-L1), we grew tumors in a syngeneic mouse model. Our data showed a significant suppression of tumor growth with the combination treatment compared to any of the single agents. Different attempts have been conducted by scientists to find novel strategies that elevate the efficacies of IO agents via the combination of epigenetic treatment with immunotherapy. However, none of those studies have yet achieved optimal results for clinical use [29]. Further studies are needed to explore novel strategies using these combinations, since epigenetic mechanisms can influence both tumor cells and immune cells, including CD4, CD8, and regulatory T-cells. Altering the epigenetic status in the tumor microenvironment has the potential to increase the sensitivities of IO agents [12].

DNA methyltransferase inhibitors encourage tumor cells to express MHCs and various tumor antigens and affect cytokine production, which facilitates tumor antigenicity [12]. Chiappinelli et al. have reported that DNA methyltransferase inhibitors induce type I IFN signaling and apoptosis in ovarian cancer cells and sensitize them to anti CTLA4 therapy in a melanoma mouse model [13]. In syngeneic mouse models of mammary carcinoma and mesothelioma, significant immune-related antitumor activity of DNA methyltransferase inhibitors was shown when combined with CTLA-4 blockade [30]. In Hodgkin lymphoma patients, a high complete response rate was observed when 5-aza was administered prior to IO agents, although the study had a small sample size and was not a randomized trial [31].

HDAC inhibitors normalize the aberrant repression of gene expression caused by HDAC and promote cell apoptosis [15]. Recently, HDAC inhibitors have been reported to increase tumor-specific antigens in cancer cells, which leads them to respond more to immunotherapy [11]. Romidepsin is a pan-HDAC inhibitor and is clinically utilized in cutaneous and peripheral T-cell lymphoma [32]. Murakami et al. demonstrated that romidepsin promotes T-cell-mediated destruction in murine melanoma cells [33]. In addition, another study showed that the combination of romidepsin and 5-fluorouracil upregulates p21 and MHC class II genes for caspase-3/7 activation in colon cancer cells [34]. Therefore, romidepsin is considered to facilitate attacks to cancer cells by the host immune system. From these insights, we hypothesized that epigenetic treatment increases the efficacy of IO agents through altering immune-related gene signatures.

In this study, the results of a microarray, TaqMan, and qRT-PCR showed an upregulation of HLA-DRA, HMOX1, IFI6, IL12A, IRF7, NFKB2, RPL3L, STAT1, STAT3, and PD-L1, whereas CSF1, CSF2, and FAS were downregulated after treatment with 5-aza and romidepsin in vitro. In a previous study, IRF7 and STAT1 were upregulated, stimulating the type I IFN pathway, after treatment with DNA methyltransferase inhibitors [13]. IRF7 is a major regulator of type I IFN production and is induced by IFNs through a STAT-dependent mechanism [17,35]. Among the upregulated molecules in our findings, IRF7 and NF-κB are activated by same signaling pathways, and they cooperate in the regulation of IFNβ to promote innate immune responses [36]. On the other hand, our results showed decreased expressions of CSF1, CSF2, and FAS following epigenetic treatment. Considering that 5-aza and romidepsin act to restore their targeted gene expressions, which are inhibited by DNA methylation and histone deacetylase, these molecules were not influenced by the treatment directly, but by some other upstream molecules. Because CSF1 and CSF2 recruit myeloid-derived suppressor cells (MDSCs) [37], a lower level of CSF1 and CSF2 with epigenetic treatment weakens the activities of MDSCs. These results indicate that epigenetic treatment works to attenuate an immunosuppressive TME, which is one of the important elements for cancer progression and therapeutic resistance; that is, epigenetic treatment has antitumor effects [38]. Meanwhile, in both in vitro and in vivo settings, STAT1, STAT3, and PD-L1 were elevated by treatment with epigenetic agents. STAT1 and STAT3 are activated by IFN-γ through JAK1/2 and upregulate the expression of PD-L1 for the recruitment of regulatory T-cells while inhibiting effector T-cell activity [38,39,40]. Therefore, treatment with epigenetic agents can induce both immunoresponsive and immunosuppressive effects on the TME, which means not necessarily favorable antitumor effects.

In the syngeneic mouse model, mice treated with the combination of epigenetic agents and anti PD-L1 drugs showed tumor growth suppression compared to untreated mice. However, there were no statistical differences between the combination treatment and either of the monotherapies. SCCVII, a murine HNSCC cell line, proliferated rapidly in cultured medium, and tumors derived from the cells also grew so fast that their volumes reached 1000 mm^3^ after about ten days from the initial stage of tumor development. The rapid growth of tumors may potentially have masked the differences between these therapeutic arms. It is considered meaningful that, in such circumstances, the combination treatment suppressed tumor growth more than the control, and there were no statistically significant differences between the control and either of the monotherapies. Considering the protein expression data in the treated tumors, PD-L1, which is upregulated by epigenetic drugs, became a target of the anti PD-L1 drug, which drew more effective antitumor effects, potentially causing a suppression of regulatory T cells and a promotion of cytotoxic T cell activation.

To sum up, the combination of 5-aza and romidepsin leads to a both immunoresponsive and immunosuppressive TME. However, the IO agent exploits the unfavorable change caused (in this case, upregulated PD-L1) to increase its efficacy. Meanwhile, the modification of the TME by an IO agent can boost the antitumor efficacies of these epigenetic drugs. That is, treatments with epigenetic agents and an IO agent interact closely to increase the therapeutic efficacy of each.

Although our study supports the current literature on the relationship between epigenetic mechanisms and IO agents, it has some limitations. The influences on other components of the TME, such as lymphocytes and stromal cells, caused by treatment with epigenetic agents were not evaluated in this study. Because epigenetic drugs can influence these non-cancerous cells in the TME, further investigations are warranted to elucidate these mechanisms. Furthermore, in the in vivo experiments conducted in this study, sample size calculation was not performed prior to experimentation. To enhance the validity and robustness of our findings, future studies should consider validating the combined therapy under conditions with appropriately determined sample sizes, potentially using smaller tumors or employing settings in which immune cells are depleted.

In conclusion, epigenetic treatment changes immune-related gene signatures to increase the efficacy of IO agents in HNSCC. Although further studies are needed, our results provide a preclinical rationale for the combination strategy of epigenetic drugs and IO agents.

## 5. Conclusions

Our study highlights the potential of a combination strategy employing both epigenetic treatment and immunotherapy in HNSCC, as the epigenetic modulation of immune-related gene signatures enhances the efficacy of IO agents. These findings provide a solid preclinical rationale for combining epigenetic drugs with IO agents to improve therapeutic outcomes in HNSCC.

## Figures and Tables

**Figure 1 cancers-17-02431-f001:**
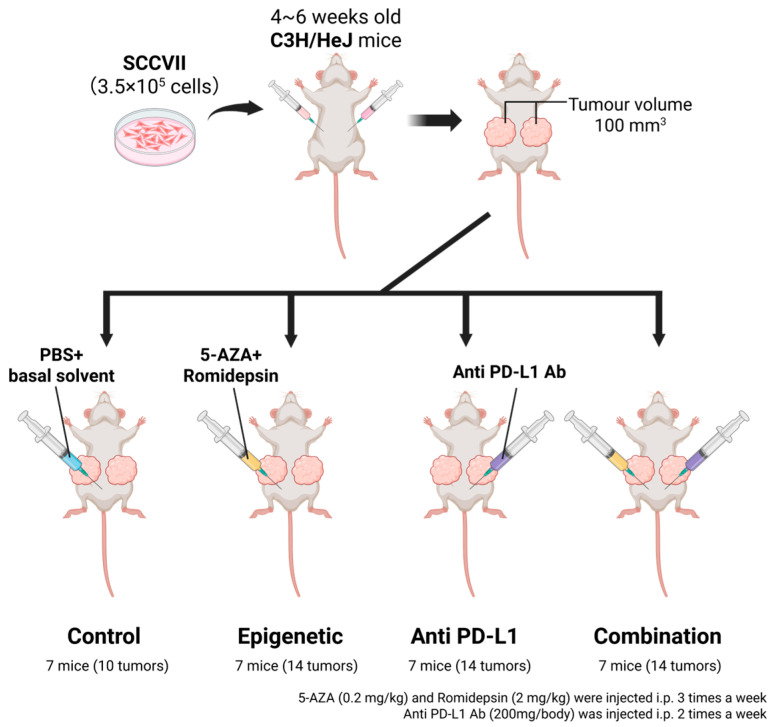
Schema of in vivo homograft therapeutic experiments [22]. SCCVII (3.5 × 10^5^ cells), that is, a murine HNSCC cell line, suspended in 100 μL of PBS was injected into both flanks of 4–6-week-old C3H/HeJ mice. When tumor volumes reached 100 mm^3^, mice were randomly assigned to four therapeutic groups (seven mice per group). 5-aza, romidepsin, and anti PD-L1 antibody were used for the treatment. 5-aza and romidepsin were administered via intraperitoneal (i.p.) injection three times a week. PD-L1 Ab was administered via i.p. injection twice a week. Therapeutic efficacy was evaluated from the percentage change in tumor volume compared to the tumor volume before treatment was started.

**Figure 2 cancers-17-02431-f002:**
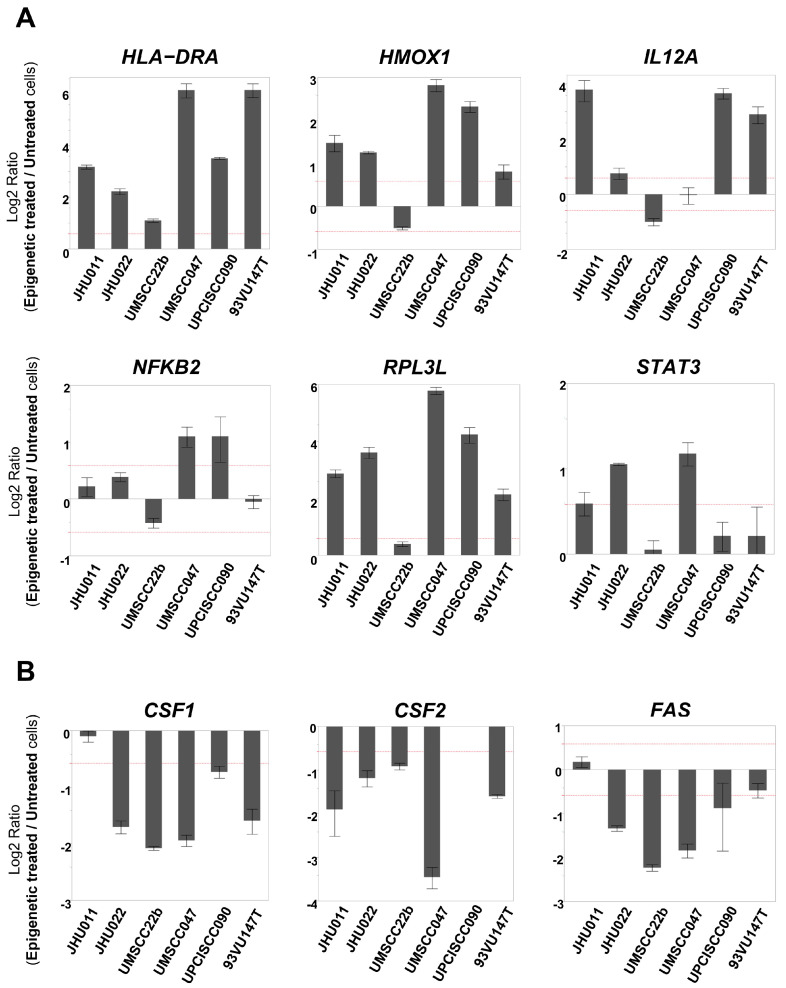
mRNA expression levels of immune-related genes that were determined to be changed in TaqMan Low Density Human Immune Array. (Each gene expression level was compared between epigenetically treated cells and untreated cells. When the fold change exceeded 1.5 (as indicated by the dotted line), it was determined as ‘increased’ or ‘decreased’). (**A**) The expression level of *HLA-DRA* was increased in all six cell lines. The expression levels of *HMOX1* and *RPL3L* were increased in five cell lines other than UMSCC22b. The expression levels of *IL12A*, *STAT3*, and *NFKB2* were increased in four, three, and two cell lines, respectively. (**B**) The expression levels of *CSF1* and *CSF2* were decreased in five cell lines, and *FAS* was decreased in four cell lines.

**Figure 3 cancers-17-02431-f003:**
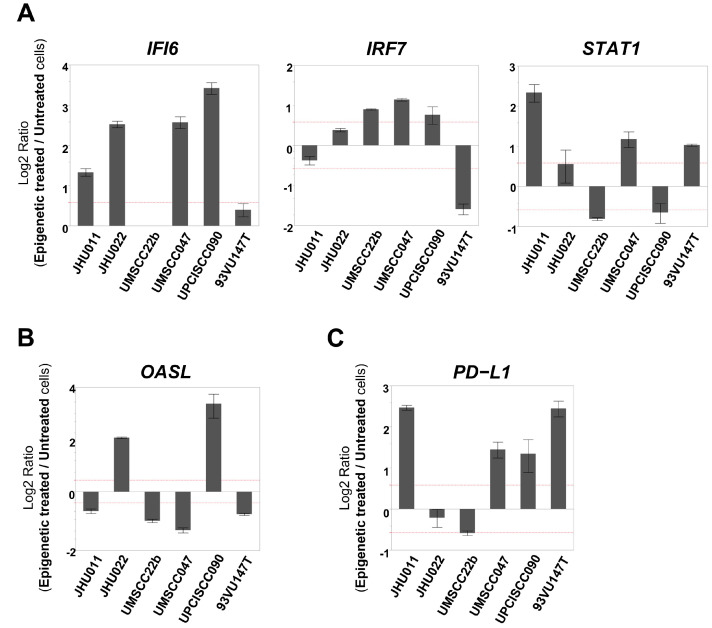
mRNA expression levels of immune-related genes that were determined to be changed in the IFN Response TaqMan custom card. (Each gene expression level was compared between epigenetically treated cells and untreated cells. When the fold change exceeded 1.5 (as indicated by the dotted line), it was determined as ‘increased’ or ‘decreased’). (**A**) *IFI6* was upregulated in four cell lines. The expression levels of *IRF7* and *STAT1* were increased in three cell lines. (**B**) The *OASL* expression level was upregulated in two cell lines and downregulated in four cell lines. (**C**) The *PD-L1* expression level was increased in four cell lines.

**Figure 4 cancers-17-02431-f004:**
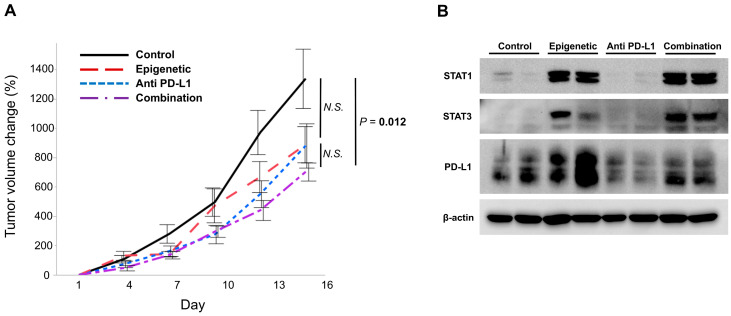
Efficacy of the combinational use of epigenetic drugs and the anti PD-L1 drug in a syngeneic mouse model. (**A**) The growth of tumors from SCCVII cells was significantly suppressed in the ‘Combination’ group compared to ‘Control’, whereas the ‘Epigenetic’ and ‘Anti PD-L1′ groups did not show statistical differences compared to ‘Control’. (**B**) When the protein expressions of STAT1, STAT3, and PD-L1 in the treated tumors were evaluated, STAT1 expression was increased in tumors in the ‘Epigenetic’ and ‘Combination’ groups, and STAT3 was increased in tumors in the ‘Epigenetic’, ‘Anti PD-L1’, and ‘Combination’ groups. PD-L1 was increased in tumors in the ‘Epigenetic’ group, and tumors in the ‘Combination’ group expressed higher PD-L1 than those in the ‘Anti PD-L1′ group but lower than those in the ‘Epigenetic’ group.

**Table 1 cancers-17-02431-t001:** Alteration of gene expressions in TaqMan Low Density Human Immune Array.

Cell Lines Genes	JHU011	JHU022	UMSCC22b	UMSCC047	UPCISCC090	93VU147T
** *HLA-DRA* **	3.32	4.69	5.04	10.68	X	X
** *HMOX1* **	1.30	2.54	1.72	7.06	9.12	1.01
** *IL12A* **	X	2.42	3.74	−1.00	9.20	15.12
** *NFKB2* **	−1.37	4.12	2.34	2.23	3.81	1.78
** *RPL3L* **	X	X	3.40	20.70	111.66	5.20
** *STAT3* **	1.50	2.19	4.02	1.16	1.44	−1.53
** *CSF1* **	−2.19	−3.10	−1.05	−7.55	X	−2.02
** *CSF2* **	−7.57	−1.76	1.88	−27.78	X	−2.45
** *FAS* **	−2.69	X	0.05	−30.70	−10.90	−2.61

Each ratio indicates the fold change in the gene expression level in epigenetically treated cells compared to untreated cells. X: Ratio was not obtained.

**Table 2 cancers-17-02431-t002:** Alteration of gene expressions in IFN Response TaqMan custom card.

Cell Lines Genes	JHU011	JHU022	UMSCC22b	UMSCC047	UPCISCC090	93VU147T
** *IFI6* **	2.49	2.46	9.92	6.95	12.81	1.16
** *IRF7* **	3.98	1.09	2.89	3.11	4.22	−2.17
** *STAT1* **	1.78	2.59	1.39	2.95	2.76	1.20
** *OASL* **	−1.81	−1.72	13.99	−3.03	90.61	−1.69

Each ratio indicates the fold change in the gene expression level in epigenetically treated cells compared to untreated cells.

## Data Availability

The original contributions presented in this study are included in the article/Appendix A. Further inquiries can be directed to the corresponding author.

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
