# Peer review of "Epigenetic Treatment Alters Immune-Related Gene Signatures to Increase the Sensitivity of Anti PD-L1 Drugs"

_cancers, 2025, doi:10.3390/cancers17152431_

Round 1

Reviewer 1 Report

Comments and Suggestions for Authors

The work is interesting, its methodology is sound, and it opens up new perspectives for the use of IO treatments.
M&M mentions that they used HPV-positive/negative cell lines to consider all presentations of this type of cancer.
However, they don't mention HPV in the introduction; this condition should be included in a short paragraph.

Fig. 1 could be improved; it is very overloaded and not easy to understand, as is Fig. 2.

Fig. 3 is not necessary 

Author Response

Reviewer: 1

Comment: The work is interesting, its methodology is sound, and it opens up new perspectives for the use of IO treatments.

Response: We sincerely appreciate the reviewer’s encouraging comments. The manuscript has been revised according to the comments.

Comment: M&M mentions that they used HPV-positive/negative cell lines to consider all presentations of this type of cancer. However, they don't mention HPV in the introduction; this condition should be included in a short paragraph.

Response: As pointed out, we added a brief explanation of HPV in HNSCC to the introduction section. (page 7, line 73- page8, line 77).

Comment: Fig. 1 could be improved; it is very overloaded and not easy to understand, as is Fig. 2.

Response: In response to reviewer’s comment, we revised the layout and text in Figures 1 and 2 (renamed Figures 2 and 3; please see next response) to make them easier to understand.

Comment: Fig. 3 is not necessary.

Response: As pointed out by the reviewer, this is not data, so it was inappropriate to show it in the results section. However, since this schema clearly shows the conditions of our in vivo experiments, we have replaced Figure 1 and moved it in the Materials & Methods section (page 6).

Reviewer 2 Report

Comments and Suggestions for Authors

The authors have summarized their findings regarding the combination of epigenetic reprogramming and PD-1 blockade. Epigenetic treatment such as 5aza could induce immune-related molecules including HLA-DR, STAT family, and PD-L1. Please find several comments to improve the merit of this paper.

  1. Immune parameters such as PD-L1 were not upregulated in all the treated cells. How to select responders to epigenetic treatment?
  2. The PD-L1 of SCCVII should be examined in in vitro with epigenetic treatments.
  3. The tumor size of combination group does not seem significantly small compared to the control group although p value is small. How many mice were used in this study? The authors are advised to test this combination in a small tumor or deplete immune cells to confirm that the combination therapy is really effective.

Author Response

Reviewer: 2

Comment 1: The authors have summarized their findings regarding the combination of epigenetic reprogramming and PD-1 blockade. Epigenetic treatment such as 5aza could induce immune-related molecules including HLA-DR, STAT family, and PD-L1. Please find several comments to improve the merit of this paper.
Response: We sincerely appreciate the reviewer’s encouraging comments. The manuscript has been revised according to the comments.

Comment 2. Immune parameters such as PD-L1 were not upregulated in all the treated cells. How to select responders to epigenetic treatment?

Response: As HNSCC is a heterogeneous cancer, it is likely that in all the treated cell lines PD-L1 will be upregulated. Furthermore, it is widely accepted that PD-L1 expression can’t be consider as a marker of IO response.

Comment 3. The PD-L1 of SCCVII should be examined in in vitro with epigenetic treatments.

Response: We thank the reviewer for this wonderful suggestion. We have plan to perform detail studies including mechanistic insight of epigenetic and IO combinations therapy response. Therefore, we will perform suggested studies in near future.

Comment 4: The tumor size of combination group does not seem significantly small compared to the control group although p value is small. How many mice were used in this study? The authors are advised to test this combination in a small tumor or deplete immune cells to confirm that the combination therapy is really effective.

Response: We appreciate this really helpful comment. We used seven mice per group (since tumors were seeded on both sides, each group had a maximum of 14 targeted tumors. However, the side of tumor that did not form or were less than 100 mm³ were excluded. The reviewers' suggestion to test this combination therapy in a small tumor or deplete immune cells is highly reasonable. We added these proposals as future experimental directions in the Discussion section (page 12, line 452-457).

Reviewer 3 Report

Comments and Suggestions for Authors
  1. Please, expand by a little the introduction by addressing any translational finding already reported in the literature about the topic.
  2. The study should clearly state and being reported according to ARRIVE guidelines
  3. Sample size calculation about mouse model?
  4. It is unclear which experiments have been made in triplicate
  5. I don't know what is this "In this section, where applicable, authors are required to disclose details of how gen- 219 Gerative artificial intelligence (GenAI) has been used in this paper (e.g., to generate text, 220 data, or graphics, or to assist in study design, data collection, analysis, or interpretation). 221The use of GenAI for superficial text editing (e.g., grammar, spelling, punctuation, and 222 formatting) does not need to be declared. 
  6. I can't clearly understand which one is the control sample. Was microarray done for non-treated cells? Are log2fc expression of which comparison? Results should be normalized against untreated models. 

Author Response

Reviewer: 3

We sincerely appreciate the reviewer’s encouraging comments. The manuscript has been revised according to the comments.

Comment 1. Please, expand by a little the introduction by addressing any translational finding already reported in the literature about the topic.

Response: We appreciate this really helpful comment. We used both HPV-positive and negative cell lines to consider all presentations of this type of cancer in this study. Therefore, we added translational findings about HPV in HNSCC to the introduction section. (page 7, line 73- page8, line 77).

Comment 2. The study should clearly state and being reported according to ARRIVE guidelines

Response: According to reviewer’s comment, we filled out and attached an ARRIVE guidelines checklist. We also revised the manuscript and Figure 1 in accordance with the checklist items (we showed the number of mice).

Comment 3. Sample size calculation about mouse model?

Response: We used seven mice per group (since tumors were seeded on both sides, each group had a maximum of 14 targeted tumors. However, the side of tumor that did not form or were less than 100 mm³ were excluded). No sample size calculation was performed. In the discussion section, we noted these limitations and added that, in order to make the results of this study more appropriate and robust, it would be desirable to conduct additional experiments with a more appropriate sample size and under more appropriate conditions. (page 12, line 452-457).

Comment 4. It is unclear which experiments have been made in triplicate

Response: All experiments were performed triplicate if not otherwise mentioned. Details of each method are described in the “Materials and Methods” section. (page 4, line 135-136, page 5, line 200 and page6 line 244-245).

Comment 5. I don't know what is this "In this section, where applicable, authors are required to disclose details of how gen- 219 Gerative artificial intelligence (GenAI) has been used in this paper (e.g., to generate text, 220 data, or graphics, or to assist in study design, data collection, analysis, or interpretation). 221The use of GenAI for superficial text editing (e.g., grammar, spelling, punctuation, and 222 formatting) does not need to be declared.

Response: We appreciate this really helpful comment. The MDPI manuscript format remained unchanged by our mistake. We deleted the text.

Comment 6. I can't clearly understand which one is the control sample. Was microarray done for non-treated cells? Are log2fc expression of which comparison? Results should be normalized against untreated models.

Response: In both microarray and qRT-PCR, we used vehicle treated cells as control samples. The log2fc expression ratios shown in the figure also represent the expression ratios of epigenetically treated cells relative to vehicle treated cells. We added descriptions of the treated cells and controls for each experiment in the Materials and Methods section (page 5, line 177-178, line 189-190, line 196- 197) and emphasized the axis labels in each graph for clarity (Figure2 and 3).

Round 2

Reviewer 2 Report

Comments and Suggestions for Authors

Although the authors have not provided additional results, they have added the future direction in the discussion.

Reviewer 3 Report

Comments and Suggestions for Authors

No further comment